# Associations between vision impairment and vision-related interventions on crash risk and driving cessation: systematic review and meta-analysis

Helen Nguyen ![ORCID],[1] Gian Luca Di Tanna,[2,3] Kristy Coxon ![ORCID],[4] Julie Brown,[2] Kerrie Ren,[1] Jacqueline Ramke ![ORCID],[5,6] Matthew J Burton,[5,7] Iris Gordon,[5] Justine H Zhang ![ORCID],[5] João Furtado,[8] Shaffi Mdala,[9] Gatera Fiston Kitema,[10] Lisa Keay[1,2]

For numbered affiliations see end of article.

**Correspondence to**
Professor Lisa Keay;
l.keay@unsw.edu.au

## ABSTRACT

**Objectives** To systematically investigate the associations between vision impairment and risk of motor vehicle crash (MVC) involvement, and evaluate vision-related interventions to reduce MVCs.

**Design** Medline (Ovid), EMBASE and Global Health electronic databases were systematically searched from inception to March 2022 for observational and interventional English-language studies. Screening, data extraction and appraisals using the Joanna Briggs Institute appraisal tools were completed by two reviewers independently. Where appropriate, measures of association were converted into risk ratios (RRs) or ORs for meta-analysis.

**Participants** Drivers of four-wheeled vehicles of all ages with no cognitive declines.

**Primary and secondary outcomes** MVC involvement (primary) and driving cessation (secondary).

**Results** 101 studies (n=778 052) were included after full-text review. 57 studies only involved older drivers (≥65 years) and 85 were in high-income settings. Heterogeneity in the data meant that most meta-analyses were underpowered as only 25 studies, further split into different groups of eye diseases and measures of vision, could be meta-analysed. The limited evidence from the meta-analyses suggests that visual field defects (four studies; RR 1.51 (95% CI 1.23, 1.85); p<0.001; $I^2$=46.79%), and contrast sensitivity (two studies; RR 1.40 (95% CI 1.08, 1.80); p=0.01, $I^2$=0.11%) and visual acuity loss (five studies; RR 1.21 (95% CI 1.02, 1.43); p=0.03, $I^2$=28.49%) may increase crash risk. The results are more inconclusive for available evidence for associations of glaucoma (five studies, RR 1.27 (95% CI 0.67, 2.42); p=0.47; $I^2$=93.48%) and cataract (two studies RR 1.15 (95% CI 0.97, 1.36); p=0.11; $I^2$=3.96%) with crashes. Driving cessation may also be linked with glaucoma (two studies; RR 1.62 (95% CI 1.20, 2.19); p<0.001, $I^2$=22.45%), age-related macular degeneration (AMD) (three studies; RR 2.21 (95% CI 1.47, 3.31); p<0.001, $I^2$=75.11%) and reduced contrast sensitivity (three studies; RR 1.30 (95% CI 1.05, 1.61); p=0.02; $I^2$=63.19%).

## STRENGTHS AND LIMITATIONS OF THIS STUDY

⇒ This is an up-to-date systematic review capturing literature on a variety of eye diseases and conditions, measures of vision such as visual acuity, contrast sensitivity, glare sensitivity and visual field, and vision-related interventions and their associations with motor vehicle crash involvement and driving cessation.

⇒ There were no geographical or age restrictions placed on the population of focus allowing the global impact of vision impairment on driving to be documented for all age groups.

⇒ Meta-analysis was limited due to heterogeneity in the outcome measures reported and the definitions of vision loss and or impairment used in each study. This heterogeneity also prohibited subgroup analyses by age and geographical location.

⇒ Only statistical heterogeneity was assessed and not clinical or methodological.

⇒ Publication bias was not assessed as there were less than 10 studies included in each meta-analysis.

Cataract surgery halved MVC risk (three studies; RR 0.55 (95% CI 0.34, 0.92); p=0.02; $I^2$=97.10). Ranibizumab injections (four randomised controlled trials) prolonged driving in persons with AMD.

**Conclusion** Impaired vision identified through a variety of measures is associated with both increased MVC involvement and cessation. Cataract surgery can reduce MVC risk. Despite literature being highly heterogeneous, this review shows that detection of vision problems and appropriate treatment are critical to road safety.

**PROSPERO registration number** CRD42020172153.

## INTRODUCTION

Globalisation and economic development have made driving one of the main modes of transport worldwide and passenger vehicle

travel is predicted to triple between 2015 and 2050.[1] Driving allows for independent mobility and enhances access to employment and education. Unfortunately, with more drivers on the roads, motor vehicle crashes (MVCs) and road traffic injuries are increasing worldwide. Approximately 1.35 million MVC-related fatalities occur each year with an additional 20–50 million people experiencing road-related injuries per annum.[2] The United Nations (UN) has therefore created targets within the Sustainable Development Goals (SDGs) which aim to halve road deaths by 2020 (target 3.6) and provide safe and sustainable transport systems for vulnerable road users (target 11.2).[3]

Driving is a common and valued activity for many adults. Driving cessation limits independent mobility and has been linked to depressive symptoms and poorer health in older adults.[4] Functional declines in vision disproportionately impact older drivers, as they have higher prevalence of poor vision and eye diseases.[5 6] Some countries have specific licensing requirements for older drivers[7]; however, variations in visual driving standards across jurisdictions have made it difficult to assess whether these standards have safety benefits.[8]

This review was completed in collaboration with the *Lancet Global Health* Commission on Global Eye Health[9] and aimed to systematically evaluate the evidence to (1) investigate the associations between vision impairment and risk of MVC involvement across the lifespan, and (2) evaluate vision-related interventions to reduce MVCs. Since risks can be mitigated by driving retirement, this review also considered driving cessation as a secondary outcome.

## METHODS

This systematic review was reported using Preferred Reporting Items for Systematic Reviews and Meta-Analyses guidelines[10] (online supplemental appendix 1) using a published protocol.[11] An electronic database search on Medline (Ovid), EMBASE and Global Health was conducted from their inception to March 2020, and then updated in March 2022, with no geographical restrictions. Online supplemental appendix 2 details the search strategy with table 1 describing the inclusion and exclusion criteria for studies.

The population of focus was drivers of four-wheeled motorised vehicles, of all ages, with no cognitive declines. Exposures of interest included eye diseases (eg, glaucoma, cataract, age-related macular degeneration (AMD), diabetic retinopathy (DR)) and conditions (eg, refractive errors), and measures of vision such as, but not limited to, visual acuity (VA) and contrast sensitivity (CS). Studies reporting on interventions focused on treatments that would improve vision. The primary outcome measure was MVC involvement identified from self-reported surveys or government/hospital administrative datasets. The secondary outcome was self-reported driving cessation. Due to the large volume of data collected, other surrogate measures of driving safety and driving performance planned in the original protocol were beyond the scope of this manuscript but will be reported in a separate systematic review.[11] Studies which used simulators or investigated self-regulatory driving behaviours (eg, night driving avoidance) through surveys were excluded.

All titles, abstracts and full texts were reviewed independently by two investigators using Covidence systematic review management software (Covidence non-profit SaaS Enterprise, Melbourne, Australia). All discrepancies were resolved via consultation with a third investigator. Similarly, data extraction was completed independently by two investigators using data extraction forms adapted from either the Joanna Briggs Institute (JBI) templates for observational and systematic review study designs, or Cochrane templates for interventional studies. Data extracted from the studies included design, participant and setting characteristics, exposure type and definition,

**Table 1** Study inclusion and exclusion criteria

| Inclusion | Exclusion |
| --- | --- |
| ► Interventional (RCTs) and observational (cohort, cross-sectional, case–control and case series) studies<br>► Systematic reviews with meta-analyses<br>► Studies on drivers of four-wheeled motorised vehicles of all ages<br>► Studies looking at the following exposures of interest: impairment in measures of vision (visual acuity, contrast sensitivity, visual field and glare sensitivity) or specific eye conditions including but not limited to glaucoma, cataracts, age-related macular degeneration, diabetic retinopathy, stereopsis disorders and colour vision deficiencies<br>► Studies on interventions such as vision screening, refractive correction, cataract surgery, anti-VEGF injections and other treatments to improve vision | ► Literature reviews and narrative systematic reviews<br>► Commentary articles, dissertations, abstracts, editorials and conference presentations<br>► Studies using simulators or investigated either self-regulatory driving behaviours (eg, night driving avoidance), or self-reported measures of driving safety<br>► To narrow the scope of the study, studies on populations with specific non-vision-related medical conditions (eg, dementia, epilepsy, stroke and history of medical events such as syncope), low vision or vision difficulties caused by other medical conditions (eg, hemianopia caused by brain damage)<br>► Studies which simulated vision impairment |

anti-VEGF, anti-vascular endothelial growth factor; RCTs, randomised controlled trials.

intervention details (if any), outcome measures and relevant effect measures.

Overall risk of bias for all included studies was assessed by two investigators independently with conflicts resolved by a third investigator. All quality assessments were conducted using the relevant JBI critical appraisal tools.[12] Each question on the relevant tools was categorised into either selection, detection, confounding, validity, performance, attrition or allocation bias by all authors. Thus, a range of biases were considered appropriate to this research question. Each study was given an overall 'score' on each question answered where a higher score represented less bias in the study design and execution. Based on how the questions were asked, a 'yes' indicated that some sort of measure to limit bias was undertaken. The final scores were used to assign each study as low, medium or high risk of bias, with lower scores indicating higher risk of bias.

### Statistical analysis

Associations between vision impairments and vision-related interventions with MVC involvement and driving cessation were summarised with appropriate HRs, risk ratios (RRs) or ORs. Narrative summaries were reported using the Synthesis Without Meta-analysis guidelines.[13] Heterogeneity across studies was assessed using $I^2$ statistic. Meta-analysis was conducted by converting all effect measures into RR or OR. Random-effects meta-analysis was only conducted on studies which presented data with the same outcomes, exposures and comparators, and which reported on associations adjusted for confounders to reduce bias. Data from case–control studies were not pooled for meta-analysis to minimise possible heterogeneity. No publication bias analysis was conducted as there were less than 10 studies in each meta-analysis. Reporting of the results was guided by the Meta-analysis of Observational Studies in Epidemiology guidelines.[14] All analyses were completed using STATA V.17.

### Patient and public involvement

Only existing published literature was looked at in this review and therefore no patient or public involvement was present during the design or execution of the review. Public participation may be sought out for future dissemination of this review.

### RESULTS

From the electronic database search, 5111 studies were identified after the removal of 2131 duplicates. After title and abstract screening, 243 studies remained for full-text review after which 142 studies were further excluded, leaving 101 studies for data extraction (figure 1).

Sixty-three studies (31 cross-sectional, 19 cohort, 12 case–control and 1 systematic review with meta-analysis) reported on MVC involvement alone, 34 (21 cross-sectional, 8 cohort, 2 case–control, 1 case series and 2 randomised controlled trials (RCTs)) on driving cessation, and 4 (1 cross-sectional, 2 cohort and 1 case–control) on both MVC and cessation. When split by geographical regions, 48 studies from high-income countries (HICs) and 15 studies from low/middle-income countries (LMICs) reported solely on MVC involvement, while all 34 studies looking at driving cessation only came from HICs. From the studies which reported on both MVC and driving cessation, only one was from an LMIC. Study breakdown according to driving outcome and vision impairment is shown in tables 2 and 3. The majority of studies (84%) were set in HICs and 57 studies (56%) focused on older adults. However, when looking at the 16 studies set in LMICs, all but 2 had an average study population age of less than 65 years. From the total 101 studies, only 13 (7 from HICs, 6 from LMICs; 12 cross-sectional,

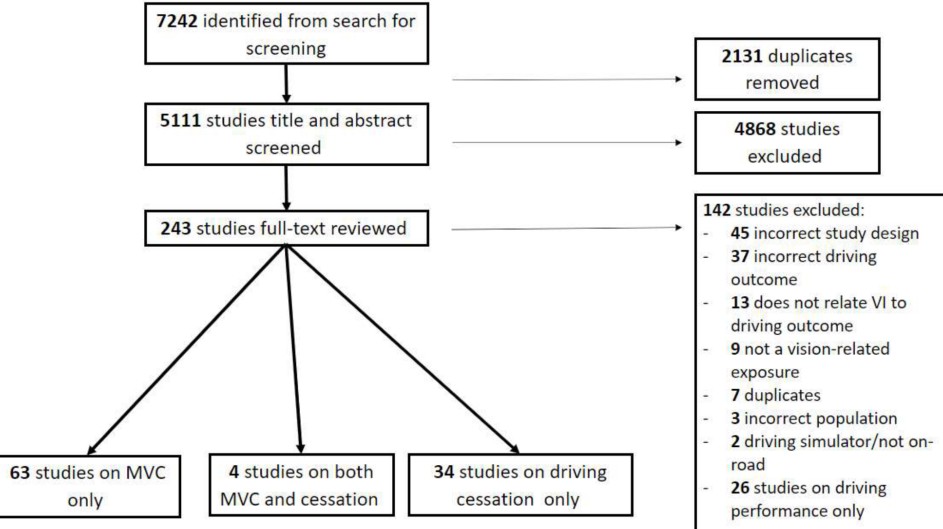

**Figure 1** Flow chart of search with papers reporting on MVC and driving cessation. MVC, motor vehicle crash; VI, vision impairment.

**Table 2** Breakdown of studies reporting on vision-related associations by outcome measure

| Driving outcome | Vision impairment | Region (HIC/LMIC) | Total no of studies |
|---|---|---|---|
| Motor vehicle crash | Glaucoma | 15 HICs; 1 LMIC | 16 |
| | Cataract | 8 HICs | 8 |
| | AMD | 6 HICs | 6 |
| | Diabetic retinopathy | 3 HICs | 3 |
| | Stereopsis impairment | 2 HICs; 3 LMICs | 5 |
| | Myopia | 2 HICs; 2 LMICs | 4 |
| | Colour blindness | 1 HICs; 7 LMICs | 8 |
| | Contrast sensitivity | 13 HICs | 13 |
| | Visual acuity | 19 HICs; 9 LMICs | 28 |
| | Glare sensitivity | 3 HICs | 3 |
| | Visual field impairment | 14 HICs; 6 LMICs | 20 |
| | Other* | 13 HICs; 6 LMICs | 19 |
| Driving cessation | Glaucoma | 12 HICs; 1 LMIC | 13 |
| | Cataract | 5 HICs | 5 |
| | AMD | 5 HICs | 5 |
| | Contrast sensitivity | 8 HICs | 8 |
| | Visual acuity | 18 HICs | 18 |
| | Glare sensitivity | 3 HICs | 3 |
| | Visual field impairment | 8 HICs | 8 |
| | Other† | 11 HICs | 11 |

*Unilateral vision impairment, general vision impairment, retinopathy, retinal detachment, poor visibility, refractive disorder, monocular vision impairment, hyperopia, amblyopia, diplopia, astigmatism, retinitis pigmentosa, stereoacuity.
†Dark adaptation, age-related maculopathy, detached retina, non-refractive vision impairment, self-reported vision loss, retinal haemorrhage, uncorrected refractive error.
AMD, age-related macular degeneration; HIC, high-income country; LMIC, low/middle-income country.

1 cohort) were categorised as high risk of bias with the rest rated as either low or medium (online supplemental appendix 3).

Raw data on studies reporting on MVCs[15–81] and driving cessation[70–73 82–115] can be found in online supplemental appendix 4A,B, respectively, with additional narrative summaries. Meta-analyses on associations are presented in online supplemental appendix 5A,B; only 25 studies could be meta-analysed. Studies were not included in the meta-analysis if different comparators were used, different driving outcomes were analysed (any MVC involvement, at-fault MVCs, injurious and non-injurious MVCs), or different cut-off points or definitions for vision impairment. For example, there were studies that looked at bilateral VA at 6/12 and worse, while there were others that looked at unilateral VA being 'poor' but without a formal definition of what 'poor' acuity meant. Studies rated as having a high bias were also excluded from the meta-analyses. Figure 2 synthesises the narrative summaries to show multiple associations of vision with MVCs and

**Table 3** Breakdown of studies reporting on a vision-related intervention by intervention type, vision impairment and outcome measure

| Intervention | Vision impairment | Driving outcome | Region (HIC/LMIC) | Studies (n) |
|---|---|---|---|---|
| Anti-VEGF injections | AMD | Driving cessation | 1 HIC | 1 |
| | Diabetic macular oedema | | 1 HIC | 1 |
| Cataract surgery | Cataract | Motor vehicle crash | 6 HICs | 6 |
| | | Driving cessation | 2 HICs | 2 |
| Corrective lenses | Refractive error | Motor vehicle crash | 1 HIC | 1 |
| Anti-glaucoma therapy | Glaucoma | Driving cessation | 1 HIC | 1 |

AMD, age-related macular degeneration; anti-VEGF, anti-vascular endothelial growth factor; HIC, high-income country; LMIC, low/middle-income country.

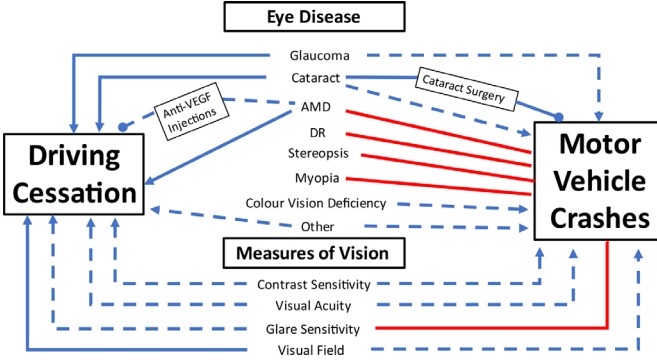

**Figure 2** Network diagram illustrating strength of association of vision impairment with motor vehicle crashes and driving cessation found by narrative summaries. Consistent associations of an increased risk of the driving outcome=solid blue line with an arrowhead; inconsistent associations of either an increased risk or no risk of the driving outcome=dashed blue line with an arrowhead; consistent associations of a decreased risk of the driving outcome=solid blue line with a closed circle; inconsistent associations of a decreased risk or no change in risk of the driving outcome=dashed blue line with a closed circle; no associations found with the driving outcome=solid red line. AMD, age-related macular degeneration; anti-VEGF, anti-vascular endothelial growth factor; DR, diabetic retinopathy.

driving cessation. From figure 2, it can be seen that associations reported for eye diseases and measures of vision function were more consistent across studies looking at cessation compared with crashes. When considering vision-related interventions, only cataract surgery was shown to improve driving by minimising crash risk. The benefits of anti-vascular endothelial growth factor (VEGF) injections on prolonging driving were more inconclusive and found to only help drivers with AMD but not diabetic macular oedema (DMO). However as a whole, the evidence from the literature on associations between vision impairment and crashes and cessation is mostly inconclusive and or mixed.

### Associations between eye diseases and conditions/measures of vision loss and MVCs

The results were mixed (16 studies, n=21 214 participants) for associations between glaucoma and MVCs.[24 30 38 41 43 45 46 52 54 65 67–72] As illustrated in online supplemental appendix 5A, meta-analyses found a glaucoma diagnosis to not increase the risk of any MVC involvement (OR 1.27 (95% CI 0.67 to 2.42); p=0.47); however, this estimate has a wide CI limiting the power to investigate this association.[24 30 38 43 72] Other studies were excluded from the meta-analysis as there was no similarity on the comparators used, how glaucoma was categorised (mild vs severe, unilateral vs bilateral) and the crash outcomes investigated (any MVC involvement, injurious vs non-injurious, at fault). Similarly, meta-analyses on three studies[24 30 43] looking at at-fault crashes also found no difference between drivers with and without glaucoma (RR 1.89 (95% CI 0.40 to 8.86); p=0.42). Increased risk was evident with more severe glaucoma.[30 38 43 46 52 65 69 70]

Out of the eight cataract studies (n=18 883) identified,[24 40 41 45 54 56 57 72] most found self-reported, physician-diagnosed cataracts did not impact the likelihood of any type of MVC involvement. Meta-analysis suggests that was no increased risk (online supplemental appendix 5A; OR 1.15 (95% CI 0.97 to 1.36); p=0.11)[24 40]; however, this was underpowered with only two studies used for analysis. At-fault crash involvement was investigated by two studies; however, only one reported significant associations.[24 56]

Meta-analysis could not be conducted on any studies looking at drivers with either AMD (five studies, n=4150)[24 41 44 64 66] or DR (three studies, n=4353)[24 45 54]; however, no studies found increased risk of MVC. No studies were meta-analysed as studies on AMD all had different comparators or different grades of AMD and MVC types, while studies on DR had different comparators and looked at different crash outcomes.

Impairments in stereopsis were not found to increase the risk of MVC involvement across the five studies identified (n=3253).[22 33 40 51 75] Meta-analysis on three studies showed no difference in crash involvement between those with and without stereopsis impairment (online supplemental appendix 5A; RR 1.03 (95% CI 0.86 to 1.23); p=0.74).[22 40 51]

Summary of studies on myopia (four studies, n=2039)[22 23 41 74] also found no increased risk of MVC involvement. A combination of two of these studies in meta-analysis (online supplemental appendix 5A) also did not find evidence of an association (OR 0.76 (95% CI 0.34 to 1.70); p=0.51),[22 74] noting limitation of sample size for concrete conclusions to be made. One study investigating persons with night myopia reported slightly more night-time MVCs in these drivers than those without night myopia (p=0.044).[23]

Colour vision deficiency and the risk of MVC involvement among commercial truck drivers were investigated in eight studies (n=7916)[15 21 22 34 51 53 59 77]; seven set in LMICs. Three studies found an association[15 51 59]; however, their results were not combined due to reliance on Ishihara plates which do not reliably diagnose colour vision deficiency.

VA (28 studies, n=39 129) was not found to be associated with crash involvement by 19 studies,[17 20 22 24 27–29 31 33–36 38 40 41 45 50–54 57 63 68 69 73 75 77 80] irrespective of crash scenario (at fault or not at fault) and severity (injurious or non-injurious). Bilateral VA 20/40 or worse may impact risk of MVCs (meta-analysis five studies; RR 1.21 (95% CI 1.02 to 1.43); p=0.03).[27 31 40 73 77] Combining two studies found no evidence for an association with 'not-at-fault' MVCs (RR 1.08 (95% CI 0.74 to 1.60); p=0.68) (online supplemental appendix 5A)[27 31]; however, there was limited power to explore associations.

Mixed results were reported from 13 studies (n=17 941) looking at any MVC involvement and reduced CS.[24 27 31 35 38 40 54 57 58 73] However, due to heterogeneity in outcome measures reported and definition of reduced CS, the meta-analysis in online supplemental appendix 5A was restricted to only two studies which found CS

to increase crash risk (RR 1.40 (95% CI 1.08 to 1.80); p=0.01).[31 79] When photopic and mesopic areas under the log CS were investigated with any and at-fault crash involvement, only lower mesopic peaks were found to be predictive.[58]

From the 20 studies (n=13 533) looking at visual field (VF) loss and crashes, heterogeneity in the definition of VF loss and the crash outcomes investigated meant that only four were meta-analysed. The results suggest an increased risk of MVC with bilateral field loss (RR 1.51 (95% CI 1.23 to 1.85); p<0.001) (online supplemental appendix 5A).[32 51 77 79] There were mixed results with 9 of 20 studies finding an increased risk,[31 32 38 42 54 73 77–79] 1 of 20 an association for a collinear dependent variable[19] and 10 of 20 a null finding.[16 17 33 34 37 51 53 59 68 69] The increased risks were found in association with severe, bilateral VF loss and field loss affecting both central and peripheral vision.

Most studies on glare sensitivity impairments (three studies, n=3191) found weak to no associations with crash risk[54 57 73]; they were unable to be meta-analysed.

Nineteen studies (n=100 167) reported on other impairments including: unilateral vision impairment,[18] general vision impairment,[21 25 28 39 41 59 61 74 76 80 81] non-DR,[41] retinal detachment,[72] other retinal disorders,[41] refractive disorder,[41] monocular vision impairment,[41 50] presbyopia,[41 74] hyperopia,[22 74] amblyopia,[18 60] diplopia,[41] astigmatism,[22 41] retinitis pigmentosa[26] and stereoacuity.[54 73] Most did not find associations with MVCs; however, one study from the USA reported increased injurious MVC involvement with impaired stereoacuity.[54] Another study in the UK reported increased MVC involvement with moderate/severe amblyopia,[41] while two other studies, one in Ethiopia[21] and the other in Bangladesh,[74] reported increased MVC involvement with self-reported bilateral visual impairment.

### Impact of vision-related interventions on MVCs

Most of the six studies (n=592 897) on cataract surgery found the risk of MVC to decrease following cataract surgery,[41 47–49 55 62] and the three studies suitable for meta-analysis estimated the risk to halve (RR 0.55 (95% CI 0.34 to 0.92); p=0.02) (online supplemental appendix 5A).[47 48 55] Greater reductions to crash risk are seen after first eye surgery compared with second eye.[47] Similarly, the risk of crashing in males post-surgery is lower than females.[49]

Corrective lenses for far and near vision refractive disorders were only investigated by one study which found no associations with crash risk.[41]

### Associations between eye diseases and conditions/measures of vision loss and driving cessation

There were 13 studies (n=21 939) investigating associations between glaucoma and the likelihood of driving cessation with estimates ranging from an increased risk of 1.3 to increased odds of 4.[70–72 87 91 92 99 100 103 109–111 113] The meta-analysis in online supplemental appendix 5B

suggests a diagnosis of glaucoma to increase the risk of driving cessation by 63% (95% CI 1.20% to 2.19%; p<0.01)[87 91]; however, this analysis only contained two studies.

Four studies (n=14 402) looked at cataract and driving cessation with three studies reporting an increased likelihood of driving cessation by over 1.5 times; none could be meta-analysed.[72 99 100 106]

From the five studies (n=6183) identified,[85 87 99 106 108] three found the presence of AMD to be predictive of driving cessation, with meta-analysis on three suitable studies reporting the overall risk of cessation to increase by 2.21 (95% CI 1.47 to 3.31; p<0.01) (online supplemental appendix 5B).[85 87 108]

Even though the 18 identified studies (n=23 712) were highly heterogeneous,[73 82 86–88 90 91 94–98 103–106 110 111] impaired or 'poor' VA was shown to increase the chances of driving cessation in most studies,[87 103 104 106 111] with better VA decreasing the risk of cessation by up to 70%.[90] The two studies looking at VA in persons with glaucoma had mixed conclusions on the effect of VA on driving cessation.[95 110]

Eight studies (n=9602) looked at the impact of CS on driving cessation.[73 88 94 96 97 103 106 111] From the studies which categorised CS as 'poor', meta-analysis found poor CS to increase the risk of cessation (RR 1.30 (95% CI 1.05 to 1.61); p=0.02) (online supplemental appendix 5B).[94 96 106] Another study reported participants who had a decline of six or more letters in their CS levels after 2 years, as measured by a Pelli-Robson chart, to have a 71% increased risk of driving cessation.[88]

VF loss and driving cessation were investigated by eight studies (n=7988),[88 94–97 103 105 111] and all but one found associations.[105] The likelihood of cessation was generally greater with bilateral and or more severe field loss.[88 94 111] One study looking at persons with bilateral glaucoma found VF loss to double the odds of cessation.[103]

Glare sensitivity (three studies, n=5577) was not found to be consistently associated with driving cessation.[88 91 110]

Eleven studies (n=12 897) looked at driving cessation with other types of vision impairment: dark adaptation,[110] age-related maculopathy,[86] retinal detachment,[85] non-refractive vision impairment,[112] general vision loss,[85 89 93 98 100 114 115] retinal haemorrhage[85] and uncorrected refractive error.[97 112] Only two studies, one reporting on retinal haemorrhage[85] and the other on non-refractive vision impairment and uncorrected refractive error,[112] found increased risk of driving cessation.

### Impact of vision-related interventions on driving cessation

There were two studies reporting the driving status of participants after anti-VEGF therapy (0.5 mg ranibizumab) from four different RCTs: MARINA (n=716; 24 months; control=sham injections) and ANCHOR (n=423; 24 months; control=photodynamic therapy (PDT)) which targeted AMD,[83] and RIDE/RISE (n=759; 24 months; control=sham injections) and RESTORE (n=345; 12 months; control=PDT) which targeted DMO.[84] By the

end of all four trials, only drivers with AMD but not DMO treated with anti-VEGF were shown to have marked differences with the control group for the number of people who continued driving from baseline (AMD: MARINA: p=0.035, ANCHOR: p=0.002; DMO: RIDE/RISE: p=0.655, RESTORE: p=0.125).

Both studies (n=1021) looking at driving status after cataract surgery reported an increase in the proportion of participants driving after successful surgery.[101 102]

There was only one study (n=240) looking at driving after anti-glaucoma therapy (pilocarpine–epinephrine)[107]; however, this is an old study and this treatment is no longer in use.

## DISCUSSION

This review synthesises diverse and complex evidence from 101 studies examining vision and its impact on MVCs and driving cessation across all ages. The majority of studies in this review focused on older adults and reported more associations between vision impairment and MVCs and or cessation compared with studies on younger populations. Research was mostly observational with few studies examining the impact of interventions to improve vision. The studies excluded from the meta-analysis tended to have mixed results regarding the associations between the vision impairment and driving outcome, whereas the studies in the meta-analyses were more consistent showing definitive associations for VA, CS and VF defects. Nonetheless, the mixed results in the narrative summaries however support the emerging idea of adding visual processing and cognitive tests alongside visual assessments to produce more predictive measures of safe driving.[116] When looking at the vision-related interventions, cataract surgery was shown to halve the risk of crashing. Others have reported that following cataract surgery, driving difficulties, such as self-reported night driving ability, reduced by 88%[117] with improvements in CS linked to these changed perceptions.[118]

Variability in the relationships between vision and MVCs may be due to several reasons. The first set of reasons surrounds how MVCs are defined and investigated in the literature. First, there are many different MVC scenarios based on the driver's role (at fault or not) and severity (injurious or non-injurious) which are not always differentiated in research studies. MVCs are also studied in a variety of ways from self-reports to analyses of large crash databases. This may cause reliability issues. For example, an American study found agreement between these two collection methods was poor when examining the total MVCs over a 3-year period.[119] Crashes can also stem from external and vehicular factors which make drawing conclusions solely based on human factors inappropriate.[120] Self-regulation, jurisdictional control on vision standards for licensing and driving cessation could all mitigate the risk of crash involvement. The second set of reasons has to do with the vision impairment themselves and the severity of the impairment. The studies which

reported increased crash risk, associated with diagnosis of an eye disease, evaluated more severe forms of the disease and worse functions of vision. Studies examining impact of a diagnosis of a disease tended to report no associations. For example, the lack of association between a diagnosis of cataract and MVC could be because the cataract is mild and is not having a significant impact on CS. A parallel review from our group has found greater defects in these measures to worsen driving performance and increase errors, which can theoretically lead to more crashes.[121] It is therefore critical to capture the severity of an eye disease and/or the actual level of vision impairment when investigating the impact of disease status on crash risk. As seen in this review, even though glaucoma, cataract and AMD had mixed or no associations with crashing, their corresponding measures of vision, mainly VF, CS and VA, respectively, were definitively associated. This may be why associations found between vision impairment and driving cessation were strong and consistent. A diagnosis of glaucoma or AMD, and poor CS were all found to increase the risk of driving cessation. Anti-VEGF injections could prolong driving for people with AMD. This is of importance as older adults greatly value independent mobility and regard driving as a vital activity for daily living.[122 123] With driving cessation linked towards multiple negative health outcomes in older adults,[4] anti-VEGF injections can have wider health benefits beyond direct impact on vision.

This review also highlights the paucity of research from LMICs despite approximately 93% of all road traffic-related deaths occurring in these countries, particularly in Africa and among young road users.[2] Despite the UN's push, most LMICs still lag behind the SDG targets on halving road traffic mortality set in the Decade of Action for Road Safety (2011–2020).[124] Previous systematic reviews point towards legislation-based interventions which modify behaviour, such as seat-belt and helmet use, to be the most effective at reducing road injuries and crash rates in LMICs.[125 126] These interventions are in line with UN recommendations for improving infrastructure, vehicle safety standards and safe road user behaviours in order to reach the targets set for SDGs 3.6 and 11.2.[127] However, there is no mention of licensing standards which need to be addressed as motorisation increases worldwide. Evidence from this global review supports vision standards for licensing to be updated, enforced and given higher priority in LMICs. Even though most LMICs do have guidelines on vision, especially for commercial drivers, it is apparent from the studies in this review that many drivers unfortunately do not satisfy these conditions. This may be because many people in LMICs lack access to eye healthcare services. The evidence for a corresponding increase in MVCs in LMICs is not well established with only one systematic review identified looking at data from these regions.[77] Though data from HICs can inform research and policy development in LMICs, increasing the evidence base from LMICs will ensure that interventions to reduce MVCs and maintain

access to driving in LMICs can be reflective of the local context.

Older drivers tend to self-regulate their driving habits by reducing their driving mileage and radius and avoiding high-risk driving situations.[128] Vision impairments have been reported to increase the likelihood of self-regulation by 19%,[129] with older drivers who self-rate their vision as 'poor' 15 times more likely to modify their driving than those who regard their vision as 'excellent'.[123] Our findings are consistent with these patterns of self-regulation, and a diagnosis of AMD or glaucoma was found in this review to be associated with driving cessation. It is likely that self-regulation is an intermediate step towards driving cessation encompassing reductions in driving frequencies and distance.[130] However, self-regulation has been reported as an insufficient compensatory measure to reduce crash risk among older drivers with a vision impairment,[131 132] which would therefore explain why glaucoma, particularly more severe glaucoma, was still linked with crashes in some studies. The relationship between crash involvement and AMD, however, was inconclusive. This may be because AMD affects central vision, thus making declines in this field easily noticeable allowing individuals to appropriately adapt their driving behaviours. Laboratory studies simulating central vision impairments show negative impacts on driving performance and safety, particularly with increasing age and distraction.[133] Further research is needed on driving patterns and behaviours of individuals with eye diseases.

Few studies, all from LMICs, in this review reported associations between colour vision deficiency and crash risk. Unfortunately, based on their high risk of bias, these studies were deemed unsuitable for meta-analysis. This does not mean, however, that their results should be dismissed. Previous simulation studies found persons with colour vision deficiency performed worse in driving simulations compared with those with normal colour vision.[134–136] However, these associations have not always been evident in studies of MVC risk.[137] This might be why recommendations proposed by the Commission Internationale de l'Eclairage, the international authority on lighting and signal lights, are for commercial drivers only.[135] Associations found in LMICs highlight issues regarding poor road infrastructure and lighting standards.[138] Further research is needed, with standardised diagnosis of colour vision deficiency and consideration of improvements to lighting and signals in the road environment in LMICs.

This review summarises global data on different eye diseases, declines in vision function and vision-related interventions, which makes the findings applicable worldwide considering motorisation and ongoing issues of vision loss, particularly in older people. There are, however, limitations which should be acknowledged. This review highlights the highly heterogeneous nature of research investigating the impact of vision on driving which unfortunately presented several methodological limitations. First, only a small number of studies could be synthesised for meta-analyses due to differences in study design. The underpowered meta-analyses meant that no absolute conclusions can be made from these results alone. It is therefore imperative that the meta-analyses results be considered alongside the narrative summaries to gain a full picture of the literature in this field. Further, this review did not consider how comorbidities, alongside vision impairment, can impact the risk of crash and driving cessation. Older adults with a vision impairment have been found to be twice as likely than those without a vision impairment to have five or more physical and/or cognitive comorbidities.[139] It is possible that the association with vision is confounded by the impact of comorbidities. Unfortunately, not all the studies included in this review reported on the comorbidities of their participants, limiting our ability to explore this possible source of bias and the extent to how this might have explained the heterogeneity of the pooled estimates via meta-regression. There were great variations in the comparator group used in each study and there were inconsistent cut-off points among studies looking at continuous measures of vision function. This heterogeneity also prevented subgroup analyses comparing younger with older age groups and geographical regions. Clinical and methodological heterogeneity could not be investigated, even though details on participant characteristics, relevant interventions and study designs were collected, due to the small number of studies included in each meta-analysis. Looking at these parameters, however, might have explained the high statistical heterogeneity in select meta-analyses. The published meta-analysis, however, was summarised narratively to ensure duplicate studies were not included in this evidence synthesis. Grey literature and non-English studies were not included which may have introduced publication bias and limited the number of studies identified from LMICs. Future research incorporating these areas may provide a clearer picture on how vision impairment is affecting global road safety.

In conclusion, this review summarises the global literature on the impact of vision and vision-related interventions on driving as part of the *Lancet Global Health* Commission on Global Eye Health. Select measures of vision impairment such as VF, VA and CS loss, and eye diseases such as glaucoma and AMD, were found to be associated with either crashes or driving cessation, while interventions such as cataract surgery and anti-VEGF injections mitigated these outcomes. However, the current literature is highly heterogeneous, and more studies are needed from LMICs to ensure what is known about vision and driving in these settings. Future studies should aim to address these issues to allow for the global context of vision impairment and driving safety to be better documented, which may assist in the achievement of the UN's SDG road safety targets.

**Author affiliations**
[1]School of Optometry and Vision Science, University of New South Wales, Sydney, New South Wales, Australia

 Nguyen H, *et al. BMJ Open* 2023;**13**:e065210. doi:10.1136/bmjopen-2022-065210

[2]George Institute for Global Health, The University of New South Wales, Sydney, New South Wales, Australia

[3]Department of Business Economics, Health and Social Care, University of Applied Sciences and Arts of Southern Switzerland, Manno, Switzerland

[4]School of Health Sciences, and the Translational Health Research Institute, Western Sydney University-Campbelltown Campus, Campbelltown, New South Wales, Australia

[5]International Centre for Eye Health, London School of Hygiene and Tropical Medicine, London, UK

[6]School of Optometry and Vision Science, University of Auckland, Auckland, New Zealand

[7]National Institute for Health Research Biomedical Research Centre for Ophthalmology, Moorfields Eye Hospital NHS Foundation Trust and UCL Institute of Ophthalmology, London, UK

[8]Division of Ophthalmology, Universidade de São Paulo Faculdade de Medicina de Ribeirão Preto, Ribeirao Preto, São Paulo, Brazil

[9]Ophthalmology Department, Queen Elizabeth Central Hospital, Blantyre, Southern Region, Malawi

[10]Ophthalmology Department, University of Rwanda College of Medicine and Health Sciences, Kigali, Rwanda

**Contributors** HN, LK and JR conceived the idea for the review. IG constructed the search. Title and abstract screening and data extraction were completed by HN, KR, JR, JHZ, JF, SM and GFK. HN drafted and revised the manuscript with suggestions from LK, JR, JB, KC, GLDT, KR, MJB, JHZ, IG, JF, SM and GFK who received the manuscript and provided feedback on the draft. LK acted as the guarantor of the manuscript. The final version of the manuscript was approved by all authors.

**Funding** HN is supported by the Australian Government Research Training Program (RTP) Scholarship (N/A). MJB is supported by the Wellcome Trust (207472/Z/17/Z). JR's appointment at the University of Auckland is funded by the Buchanan Charitable Foundation (N/A), New Zealand. The *Lancet Global Health* Commission on Global Eye Health was supported by grants from the Queen Elizabeth Diamond Jubilee Trust (N/A), Moorfields Eye Charity (GR001061), NIHR Moorfields Biomedical Research Centre (N/A), the Wellcome Trust (20190426_PH2), Sightsavers (N/A), the Fred Hollows Foundation (N/A), SEVA Foundation (N/A), British Council for the Prevention of Blindness (N/A) and the Christian Blind Mission (N/A).

**Competing interests** None declared.

**Patient and public involvement** Patients and/or the public were not involved in the design, or conduct, or reporting, or dissemination plans of this research.

**Patient consent for publication** Not required.

**Ethics approval** Not applicable.

**Provenance and peer review** Not commissioned; externally peer reviewed.

**Data availability statement** All data relevant to the study are included in the article or uploaded as supplemental information.

**ORCID iDs**
Helen Nguyen http://orcid.org/0000-0002-5285-1507
Kristy Coxon http://orcid.org/0000-0003-4820-0397
Jacqueline Ramke http://orcid.org/0000-0002-5764-1306
Justine H Zhang http://orcid.org/0000-0001-8385-2003

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
