## [Reviewer comments · BMJ Open]

ARTICLE DETAILS

TITLE (PROVISIONAL)	Associations between vision impairment and vision-related interventions on crash risk and driving cessation: systematic review and meta-analysis
AUTHORS	Nguyen, Helen; Di Tanna, Gian Luca; Coxon, Kristy; Brown, Julie; Ren, Kerrie; Ramke, Jacqueline; Burton, Matthew J; Gordon, Iris; Zhang, Justine; Furtado, João; Mdala, Shaffi; Kitema, Gatera Fiston; Keay, Lisa

VERSION 1 – REVIEW

REVIEWER	Swan, Garrett Cubic Corp
REVIEW RETURNED	16-Jul-2022

GENERAL COMMENTS	The authors performed a meta-analysis examining the associations between vision impairment and motor vehicle crashes (any crash and at-fault crashes) and between vision-related interventions (e.g., cataract surgery) and reducing motor vehicle crashes. Given the variety and complexity of vision impairments and the study of motor vehicle crashes, updated meta-analyses, such as this manuscript, provide valuable insights. I found this meta analysis to be well written, concise, and containing interesting and relevant details. I think the inclusion of LMIC studies is also a valuable contribution. I only have a few comments. Why were so many studies omitted from the meta-analysis (almost 75% were omitted - from 106 to 25)? On page 8 around line 26-27, I think it would be helpful to generally describe why (i.e., due to bias, lack of details, etc). I really like Figure 3, but it would help if there were indications of the direction of effect for the consistent associations, perhaps by changing the shape of the arrowhead or having a marker at the start that differs (e.g., circle vs square). In Table 1, add the "Simulated Impairment" to the list of excluded studies, given that many studies use simulated levels of impairment to homogenize vision deficits (e.g., Higgins, Wood, Tait, 1998; Higgins, Wood 2005; Wood, Troutbeck 1994; etc). I would like to see more discussion about the observed differences in the associations found in the narrative summaries vs the meta-analysis. Is it merely that the narrative summaries consider different studies / interpretations than the meta-analyses? And as
--

	a followup, do the findings from the meta-analysis also suggest adding other tests (e.g., cognitive tests)? There is obviously going to be overlap between Eye Disease and Measures of Vision. Given that this distinction was made in the results, I think it would be helpful to elaborate on their independent (if there are any) contributions to motor vehicle crash risk. For example, I find it interesting that VA and CS both are associated with motor vehicle crash, yet cataracts have a mixed association.
--	---

REVIEWER	Yuki, Kenya Keio University School of Medicine Graduate School of Medicine, Ophthalmology
REVIEW RETURNED	21-Jul-2022

GENERAL COMMENTS	The authors investigated the associations between vision impairment and risk of motor vehicle crash involvement. I have read this manuscript with interest. This manuscript have enough novelty and importance to be published in this journal.
---

REVIEWER	Isawumi , M A
REVIEW RETURNED	12-Aug-2022

GENERAL COMMENTS	This is a well thought out, investigated and achieved research results. It is an impactful topic of public health interest. It also stirs readers the thought of further suggested research especially in the LMIC. it is important to science community and the general populace
---

REVIEWER	Schuster, Alexander Mannheim Institute of Public Health, Social and Preventive Medicine, Medical Faculty, Heidelberg University
REVIEW RETURNED	05-Nov-2022

GENERAL COMMENTS	Nguyen et al. present a well-written manuscript describing the association between visual impairment and vision-related interventions on crash risk and driving cessation using a meta-analytical approach. The used methods are appropriate, the clinical impact is relevant. Nevertheless, the underlying studies show strong methodological heterogeneity, as a consequence only some studies were pooled resulting in a potential risk of bias. Specific comments: Abstract: I assume that the authors mean „visual field defect” in the results part. Methods: Please report the type of software being used for statistics. Results: it is unclear which studies were chosen and which studies were excluded for the meta-analysis in the separate aspects of MVCs and driving cessation. Please further explain (i.e. page 8 line 42: five suitable studies from 16 identified studies were only used for the association between glaucoma and MVCs.) and report the corresponding reasons. Discussion: As the authors showed, there is inconsistent evidence that some eye-diseases might lead to MVCs and others not, while
--

	there is relatively good evidence for the different aspects of visual dysfunction (VA, visual field defects, etc.). A further approach incorporating both eye diseases and the functional status might show which are the main visual function aspects for MVCs in the different eye diseases (i.e. visual field defect in glaucoma, while in AMD it might be visual acuity). Please further discuss.
--	---

REVIEWER	Murdoch, Ian UCL, Institute of Ophthalmology
REVIEW RETURNED	21-Nov-2022

GENERAL COMMENTS	This review is an amazing feat. It is very readable and presents a summary of a massive and hugely heterogenous literature on visual parameters (disease and function) in relation to motor vehicle driving and crashes. I have very few comments: 1 The final paragraph in methods is confusing. The scoring system is unclear. It seems there was a binary outcome of yes or no/unknown/not applicable for a long list of mostly negative but also positive attributes. With a score of 1 for yes, I am confused how lower scores represent a higher risk of bias, and in which direction scoring was made on, for example, validity? 2 Whilst the concept of figure 3 is very appealing I find the figure wholly inadequate and in danger of misinterpretation. Positive and negative effects are indicated with the same continuous line (eg cataract extraction and visual field loss). Factors found to have no relation still have a dotted line (eg stereopsis and myopia). Either this figure should be withdrawn, or it should receive more attention to convey the findings. I appreciate the narrative summaries have been added here but if these are so much at odds with the other analysis presented then this merits elaboration in both results and discussion. 3 Page 14 line 7 is probably missing the word 'be'?could not be investigated.....
---

REVIEWER	Brophy, James McGill University
REVIEW RETURNED	02-Feb-2023

GENERAL COMMENTS	General comments: I have been asked specifically to review the statistical aspects of this study as the substantive material is outside my clinical expertise in cardiovascular health and disease. 1. According to the manuscript "The results were mixed (16 studies, n=21,214 participants) for associations between glaucoma and MVCs. As illustrated in Figure 2a, meta-analyses on five suitable studies". There is no explanation as to how the five "suitable" studies were identified. Such an unjustified selection would seem to undercut the concept of this being a systematic review and meta-analysis. 2. it is also reported "...found a glaucoma diagnosis did not significantly increase the risk of any MVC involvement (OR 1.27 (95% CI 0.67- 2.42); p=0.47)." I don't believe this statement correctly interprets the data. The data are compatible with a 33% decrease in MVC and a 142% increase. As both these limits are certainly clinically important concluding "no significant increase" seems erroneous. Rather the conclusion would appear to be the included studies are insufficient (underpowered) to draw any meaningful conclusions. 3. There are numerous other examples where the authors confuse "absence of evidence with evidence of absence" (e.g. "showing no
--

	overall increased risk (Figure 2a; two studies; OR 1.15 (95% CI 0.97-1.36)” – this is only true if one accepts that a possible 36% increase in risk is of no clinical importance). 4. A similar example is “found no increased risk of MVC involvement which was supported by the meta-analysis of two studies in Figure 2a (OR 0.76 (95% CI 0.34-1.70; p=0.51).” Is a possible 66% decrease or 70% increase truly of no interest. Again the conclusion is not there is no increased risk but rather that the evidence is insufficient. I believe these misinterpretations are present throughout the manuscript. 5. “Meta-analysis could not be conducted on any studies looking at drivers with either AMD (five studies, n=4150) or diabetic retinopathy (3 studies, n=4353)”. No justification is provided for this statement. 6. The paper states “...visual acuity loss (5 studies; RR 1.21 (95%CI 1.02, 1.43); p=0.03, I²=28.49%) but not glaucoma (five studies, RR 1.27 (95%CI 0.67-2.42); p=0.47; I²=93.48%)...increased crash risk”. Apparently, these conclusions are strictly based on a significant p value in the first case and a non-significant one in the second case. However the point estimate for increased risk is virtually identical in the 2 cases and it is quite incorrect to make a comparison on their respective p values. (see Gelman A. The Difference Between “Significant” and “Not Significant” is not Itself Statistically Significant” (The American Statistician, November 2006, Vol. 60, No. 4)) Although only my statistical opinion was requested  1. I am a bit mystified by the statement “This review is part of the Lancet Global Health Commission on Global Eye Health”. 2. The authors report that 1 systematic review was included in their review. Can the authors assure the readers that the studies in the MA were not also included among the 101 individual studies assessed, otherwise would not double counting exist? 3. The authors recommend more studies in LMIC but I do wonder if this is really a research priority for these countries with limited resources and numerous other health and safety issues. Why would one expect associations between visual disturbances and motor vehicle crashes to vary between HIC and LMIC?
--	---

VERSION 1 – AUTHOR RESPONSE

Reviewer: 1

Comments to the Author:

The authors performed a meta-analysis examining the associations between vision impairment and motor vehicle crashes (any crash and at-fault crashes) and between vision-related interventions (e.g., cataract surgery) and reducing motor vehicle crashes. Given the variety and complexity of vision impairments and the study of motor vehicle crashes, updated meta-analyses, such as this manuscript, provide valuable insights. I found this meta analysis to be well written, concise, and containing interesting and relevant details. I think the inclusion of LMIC studies is also a valuable contribution.

I only have a few comments.

Why were so many studies omitted from the meta-analysis (almost 75% were omitted - from 106 to 25)? On page 8 around line 26-27, I think it would be helpful to generally describe why (i.e., due to bias, lack of details, etc).

Thank you for your comment. The reasons as to why so many studies were excluded has been clarified in the “Results” section starting from page 8:

“Meta-analyses on associations are presented in *Appendix 5a-b*; only 25 studies could be meta-analysed. Studies were not included in the meta-analysis if different comparators were used, different driving outcomes were analysed (any MVC involvement, at-fault MVCs, injurious and non-injurious MVCs), or different cut-off points or definitions for vision impairment. For example, there were studies that looked at bilateral VA at 6/12 and worse while there were others that looked at unilateral VA being “poor” but without a formal definition of what “poor” acuity meant. Studies rated as having a high bias were also excluded from the meta-analyses. Figure 2 synthesises the narrative summaries...”

I really like Figure 3, but it would help if there were indications of the direction of effect for the consistent associations, perhaps by changing the shape of the arrowhead or having a marker at the start that differs (e.g., circle vs square).

Thank you for the suggestion, Figure 3 has been amended to better explain the direction of effect and the legend has been amended. The explanation and legend for Figure 3 has also been amended in the “Results” starting from page 8. Please note that Figure 3 has now been changed to Figure 2 in the manuscript.

“From Figure 2 it can be seen that associations reported for eye diseases and measures of vision function were more consistent across studies looking at cessation compared to crashes.”

Figure 2 Legend:

Figure 2 Network diagram illustrating strength of association of vision impairment on motor vehicle crashes (MVCs) and driving cessation found by narrative summaries only
Consistent associations of an increased risk of the driving outcome = solid blue line with an arrowhead
Inconsistent associations of either an increased risk or no risk of the driving outcome = dashed blue line with an arrowhead
Consistent associations of a decreased risk of the driving outcome = solid blue line with a closed circle
Inconsistent associations of a decreased risk or no change in risk of the driving outcome = dashed blue line with a closed circle
No associations found with the driving outcome = solid red line

In Table 1, add the “Simulated Impairment” to the list of excluded studies, given that many studies use simulated levels of impairment to homogenize vision deficits (e.g., Higgins, Wood, Tait, 1998; Higgins, Wood 2005; Wood, Troutbeck 1994; etc).

Table 1 has been amended as suggested with the “exclusion” side now containing:

“Studies which simulated vision impairment”

I would like to see more discussion about the observed differences in the associations found in the narrative summaries vs the meta-analysis. Is it merely that the narrative summaries consider different studies / interpretations than the meta-analyses? And as a followup, do the findings from the meta-analysis also suggest adding other tests (e.g., cognitive tests)?

A discussion of the narrative summaries vs. meta-analyses has been added to the results. Such differences are mentioned in the first paragraph of the “Discussion” section page 11:

“Research was mostly observational with a few studies examining the impact of interventions to improve vision. The studies excluded from the meta-analysis tended to have mixed results regarding the associations between the vision impairment and driving outcome, whereas the studies in the meta-analyses were more consistent showing definitive associations for VA, CS and VF defects. Nonetheless, the mixed result in the narrative summaries support the emerging idea of adding visual

and processing and cognitive tests alongside visual assessments to produce more predictive measures of safe driving. When looking at the vision-related interventions, cataract surgery was ...

There is obviously going to be overlap between Eye Disease and Measures of Vision. Given that this distinction was made in the results, I think it would be helpful to elaborate on their independent (if there are any) contributions to motor vehicle crash risk. For example, I find it interesting that VA and CS both are associated with motor vehicle crash, yet cataracts have a mixed association.

Thank you for pointing this out. Additional discussion has been added to the 2nd paragraph of the "Discussion" page 11:

"Variability in the relationships between vision and MVCs may be due to several reasons. The first set of reasons surround how MVCs are defined and investigated in the literature. Firstly, there are many different MVC scenarios....mitigate the risk of crash involvement. The second set of reasons have to do with the vision impairment themselves and the severity of the impairment. The studies which reported increase crash risk, associated with diagnosis of an eye disease, evaluated more severe forms of the disease and worse functions of vision. Studies examining impact of a diagnosis of a disease tended to report no associations. For example the lack of association between a diagnosis of cataract and MVC could be because the cataract is mild and is not having a significant impact on CS. A parallel review from our group has found greater defects in these measures of vision to worsen driving performance and increase errors, which can theoretically lead to more crashes.(121) It is therefore critical to capture the severity of an eye disease and/or the actual level of vision impairment when impact of disease status on crash risk . As seen in this review, even though glaucoma, cataract and AMD had mixed or no associations with crashing, their corresponding measures of vision, mainly VF, CS and VA respectively, were definitively associated.

References

121. Nguyen H, Luca Di Tanna G, Coxon K, Brown J, Ren K, Ramke J, et al. Associations between vision impairment and driving performance and the effectiveness of vision-related interventions: A systematic review. *Transportation Research Interdisciplinary Perspectives*. 2023;17:100753.

Reviewer: 2

Comments to the Author:

The authors investigated the associations between vision impairment and risk of motor vehicle crash involvement. I have read this manuscript with interest. This manuscript have enough novelty and importance to be published in this journal.

Reviewer: 3

Comments to the Author:

This is a well thought out, investigated and achieved research results. It is an impactful topic of public health interest. It also stirs readers the thought of further suggested research especially in the LMIC. it is important to science community and the general populace

Reviewer: 4

Dr. Alexander Schuster, Mannheim Institute of Public Health, Social and Preventive Medicine

Comments to the Author:

Nguyen et al. present a well-written manuscript describing the association between visual impairment and vision-related interventions on crash risk and driving cessation using a met-analytical approach. The used methods are appropriate, the clinical impact is relevant. Nevertheless, the underlying studies show strong methodological heterogeneity, as a consequence only some studies were pooled resulting in a potential risk of bias.

Specific comments:

Abstract: I assume that the authors mean „visual field defect“ in the results part.

This has been amended in the “Abstract”:

“Visual field defects (XXX), and contrast sensitivity (XXX) and visual acuity loss(XXX) ...”

Methods: Please report the type of software being used for statistics.

The statistical software used now been added to the “Statistical Analysis” section of the “Methods” page 6:

“All analyses were completed using STATA v17.”.

Results: it is unclear which studies were chosen and which studies were excluded for the meta-analysis in the separate aspects of MVCs and driving cessation. Please further explain (i.e. page 8 line 42: five suitable studies from 16 identified studies were only used for the association between glaucoma and MVCs.) and report the corresponding reasons.

Thank you for your comment. Further explanations on why the studies were excluded have now been added to “Results” section starting from page 8:

“Meta-analyses on associations are presented in *Appendix 5a-b*; only 25 studies could be meta-analysed. Studies were not included in the meta-analysis if different comparators used, different driving outcomes analysed (any MVC involvement, at-fault MVCs, injurious and non-injurious MVCs), or different cut-off points or definitions for vision impairment. For example, there were studies that looked at bilateral VA at 6/12 and worse while there were others that looked at unilateral VA being “poor” but without a formal definition of what “poor” acuity meant. Studies rated as having a high bias were also excluded from the meta-analyses. Figure 2 synthesises the narrative summaries...”

This has also been clarified throughout the “Results” where relevant:

“The results were mixed (16 studies, XXX) ... Other studies were excluded from the meta-analysis as different comparators were used, glaucoma was defined differently (mild vs severe, unilateral vs. bilateral) and the crash outcomes were different (any MVC involvement, injurious vs. non-injurious, at-fault).”

Discussion: As the authors showed, there is inconsistent evidence that some eye-diseases might lead to MVCs and others not, while there is relatively good evidence for the different aspects of visual dysfunction (VA, visual field defects, etc.). A further approach incorporating both eye diseases and the functional status might show which are the main visual function aspects for MVCs in the different eye diseases (i.e. visual field defect in glaucoma, while in AMD it might be visual acuity). Please further discuss.

Thank you for your comment. This point has been further discussed in the second paragraph of the “Discussion” page 11:

“The studies which reported increase crash risk, associated with a diagnosis of an eye disease, evaluated more severe forms of the disease and worse functions of vision. Studies examining impact of a diagnosis of a disease tended to report no associations. For example the lack of association between a diagnosis of cataract and MVC could be because the cataract is mild and is not having a significant impact on CS. A parallel review from our group has found greater defects in these measures of vision to worsen driving performance and increase errors, which can theoretically lead to more crashes.(121) It is therefore critical to capture the severity of an eye disease and/or the actual level of vision impairment when impact of disease status on crash risk . As seen in this review, even though glaucoma, cataract and AMD had mixed or no associations with crashing, their corresponding measures of vision, mainly VF, CS and VA respectively, were definitively associated.”

References

121. Nguyen H, Luca Di Tanna G, Coxon K, Brown J, Ren K, Ramke J, et al. Associations between vision impairment and driving performance and the effectiveness of vision-related interventions: A systematic review. *Transportation Research Interdisciplinary Perspectives*. 2023;17:100753.

Reviewer: 5
Dr. Ian Murdoch, UCL

Comments to the Author:

This review is an amazing feat. It is very readable and presents a summary of a massive and hugely heterogenous literature on visual parameters (disease and function) in relation to motor vehicle driving and crashes. I have very few comments:

1 The final paragraph in methods is confusing. The scoring system is unclear. It seems there was a binary outcome of yes or no/unknown/not applicable for a long list of mostly negative but also positive attributes. With a score of 1 for yes, I am confused how lower scores represent a higher risk of bias, and in which direction scoring was made on, for example, validity?

Thank you for pointing this out. This section has been edited for clarity as follows:

“Each question on the tools were categorised into either selection, detection, confounding, validity, performance, attrition, or allocation bias by all authors. **Thus, a range of biases were considered appropriate to this research question. Each study was given an overall “score” on each question answered where a higher score represented less bias in the study design and execution. Based on how the questions were asked, a “yes” indicated that some sort of measure to limit bias was undertaken.** The final scores were used to assign each study as low, medium or high risk of bias, with lower scores indicating higher risk of bias.

2 Whilst the concept of figure 3 is very appealing I find the figure wholly inadequate and in danger of misinterpretation. Positive and negative effects are indicated with the same continuous line (eg cataract extraction and visual field loss). Factors found to have no relation still have a dotted line (eg stereopsis and myopia). Either this figure should be withdrawn, or it should receive more attention to convey the findings. I appreciate the narrative summaries have been added here but if these are so much at odds with the other analysis presented then this merits elaboration in both results and discussion.

Thank you for the comment. Figure 2 has been re-drawn to make the associations and their directions clearer. The data presented in this Figure has also been restricted to the narrative summaries only to minimise confusion when looking at the results (amended in the “Results” section page 8). Please also not that the Figure 3 has been changed to Figure 2 in the manuscript. Explanation as to why some of the narrative summaries differed with the meta-analyses has also been added to the 1st paragraph of the “Discussion” section page 11.

New Legend of Figure 2:

Figure 2 Network diagram illustrating strength of association of vision impairment on motor vehicle crashes (MVCs) and driving cessation found by narrative summaries only

- Consistent associations of an increased risk of the driving outcome = solid blue line with an arrowhead
- Inconsistent associations of either an increased risk or no risk of the driving outcome = dashed blue line with an arrowhead
- Consistent associations of a decreased risk of the driving outcome = solid blue line with a closed circle
- Inconsistent associations of a decreased risk or no change in risk of the driving outcome = dashed blue line with a closed circle
- No associations found with the driving outcome = solid red line

Explanation as to why some of the narrative summaries differed with the meta-analyses has also been added to the 1st paragraph of the “Discussion” section page 11.

“The studies excluded from the meta-analysis tended to have mixed results regarding the associations between the vision impairment and driving outcome, whereas the studies in the meta-analyses were more consistent showing definitive associations for VA, CS and VF defects.

3 Page 14 line 7 is probably missing the word ‘be’?could not be investigated.....

The line has been amended:

“Clinical and methodological heterogeneity could not be investigated...”

Reviewer: 6

Comments to the Author:

Title: Associations between vision impairment and vision-related interventions on crash risk and driving cessation: systematic review and meta-analysis

Summary:

General comments: I have been asked specifically to review the statistical aspects of this study as the substantive material is outside my clinical expertise in cardiovascular health and disease.

1. According to the manuscript “The results were mixed (16 studies, n=21,214 participants) for associations between glaucoma and MVCs. As illustrated in Figure 2a, meta-analyses on five suitable studies”. There is no explanation as to how the five “suitable” studies were identified. Such an unjustified selection would seem to undercut the concept of this being a systematic review and meta-analysis.

Thank you for your comment. The reason why so few studies were deemed “suitable” for meta-analysis has been clarified in the “Results” section, page 8. Please note that Figure 2a as now been changed to Appendix 5a-b.

“Meta-analyses on associations are presented in *Appendix 5a-b*; only 25 studies could be meta-analysed. Studies were not included in the meta-analysis if different comparators used, different driving outcomes analysed (any MVC involvement, at-fault MVCs, injurious and non-injurious MVCs), or different cut-off points or definitions for vision impairment. For example, there were studies that looked at bilateral VA at 6/12 and worse while there were others that looked at unilateral VA being “poor” but without a formal definition of what “poor” acuity meant. Studies rated as having a high bias were also excluded from the meta-analyses. Figure 2 synthesises the narrative summaries...”

2. it is also reported “...found a glaucoma diagnosis did not significantly increase the risk of any MVC involvement (OR 1.27 (95% CI 0.67- 2.42); p=0.47).” I don’t believe this statement correctly interprets the data. The data are compatible with a 33% decrease in MVC and a 142% increase. As both these limits are certainly clinically important concluding “no significant increase” seems erroneous. Rather the conclusion would appear to be the included studies are insufficient (underpowered) to draw any meaningful conclusions.

Thank you for pointing this out. The authors now understand where issues in interpreting the data have arisen. As most of the meta-analyses in this review would have been underpowered, the “Results” section (page 8-10) has been amended as follows:

“As illustrated in Appendix 5a, meta-analysis found a glaucoma diagnosis to not increase the risk of any MVC involvement (OR XXX) however this estimate had a wide confidence interval limiting the power to investigate this association. Other studies were excluded from the meta-analysis as there was no similarity on the comparators used, how glaucoma was categorised

(mild vs severe, unilateral vs. bilateral) and the crash outcomes investigated (any MVC involvement, injurious vs. non-injurious, at-fault). Similarly, meta-analyses on three...

“Out of the eight cataract studies (n=18,883) identified, most found self-reported, physician-diagnosed cataracts did not impact the likelihood of any type of MVC involvement. Meta-analysis suggests that there was no increased risk (*Appendix 5a*; OR 1.15 (95% CI 0.97-1.36); p=0.11) however this was underpowered with only two studies used for analysis. At-fault crash involvement was investigated by two studies however only one reported significant associations.”

“Summary of studies on myopia (four studies, n= 2039) also found no increased risk of MVC involvement. A combination of 2 of these studies in meta-analysis (*Appendix 5a*) also did not find evidence for an association (OR 0.76 (95% CI 0.34-1.70; p=0.51), noting limitations of sample size for concrete conclusions to be made. One study investigating persons with night myopia...”

“Bilateral VA 20/40 or worse may impact risk of MVCs (meta-analysis five studies; RR 1.21 (95% CI 1.02-1.43); p= 0.03). Combining two studies found no evidence for an association with ‘not at-fault’ MVCs (two studies; RR 1.08 (95% CI 0.74-1.60); p= 0.68) (*Appendix 5a*), however there was limited power to explore this association.”

“From the 20 studies (n=13,533) looking at visual field (VF) loss and crashes, heterogeneity in the definition of VF loss and the crash outcomes investigated meant that only four studies were meta-analysed. The results suggest an increased risk of MVC with bilateral field loss (RR 1.51 (95% CI 1.23-1.85); p< 0.001) (*Appendix 5a*). There were mixed results with 9/20 studies....”

“The meta-analysis in *Appendix 5b* suggests a diagnosis of glaucoma to increase the risk of driving cessation by 63% (95% CI 1.20 - 2.19; p< 0.01), however this analysis only contained two studies.”

“Eight studies(n=9602) looked at the impact of CS...From the studies which categorised contrast sensitivity as “poor”, meta-analysis on three studies found poor CS to increase the risk of cessation (RR 1.30 (95% CI 1.05-1.61); p=0.02) (*Appendix 5b*). Another study reported participants...”

Further explanation has also been added to the “limitations” paragraph of the “Discussion”, page 12:

“There are, however, limitations which should be acknowledged. This review highlights the highly heterogeneous nature of research investigating the impact of vision on driving which unfortunately presented several methodological limitations. Firstly, only a small number of studies could be synthesized for meta-analyses due to differences in study design. The underpowered meta-analyses meant that no absolute conclusions can be made from these results alone. It is therefore imperative that the meta-analyses results be considered alongside the narrative summaries to gain a full picture of the literature in this field. There were great variations in the comparator group...”

3. There are numerous other examples where the authors confuse “absence of evidence with evidence of absence” (e.g. “showing no overall increased risk (Figure 2a; two studies; OR 1.15 (95% CI 0.97-1.36)” – this is only true if one accepts that a possible 36% increase in risk is of no clinical importance).

Thank you for this comment. This has been amended as highlighted in the previous response. Please note that Figure 2a and now been changed to Appendix 5a-b.

4. A similar example is “found no increased risk of MVC involvement which was supported by the meta-analysis of two studies in Figure 2a (OR 0.76 (95% CI 0.34-1.70; p=0.51).” Is a possible 66% decrease or 70% increase truly of no interest. Again the conclusion is not there is no increased risk but rather that the evidence is insufficient. I believe these misinterpretations are present throughout the manuscript.

The “Results” section has been amended as above to reflect the appropriate interpretation. Please note that Figure 2a and now been changed to Appendix 5a-b.

5. “Meta-analysis could not be conducted on any studies looking at drivers with either AMD (five studies, n=4150) or diabetic retinopathy (3 studies, n=4353)”. No justification is provided for this statement.

Justification for why so many studies were not included in the meta-analysis has been added as above (query 1). This has been re-explained in the relevant section pointed out as follows:

“Meta-analyses could not be conducted on any studies.... No studies were meta-analysed as studies on AMD all had different comparators or different grades of AMD and MVC type, while studies on DR had different comparators and looked at different crash outcomes.”

6. The paper states “...visual acuity loss (5 studies; RR 1.21 (95%CI 1.02, 1.43); p=0.03, I²=28.49%) but not glaucoma (five studies, RR 1.27 (95%CI 0.67-2.42); p=0.47; I²=93.48%)...increased crash risk”. Apparently, these conclusions are strictly based on a significant p value in the first case and a non-significant one in the second case. However the point estimate for increased risk is virtually identical in the 2 cases and it is quite incorrect to make a comparison on their respective p values. (see Gelman A. The Difference Between “Significant” and “Not Significant” is not Itself Statistically Significant” (The American Statistician, November 2006, Vol. 60, No. 4))

Thank you for your guidance on this. Alongside changes to the “Results” section as above, the “Abstract” has also been amended to reflect the more appropriate interpretation:

“Results: 101 studies (n=778,052) were included after full-text review. 57 studies only involved older drivers (≥65 years) and 85 were in high-income settings. Heterogeneity in the data meant that most meta-analyses were underpowered as only 25 studies, further split into different groups of eye diseases and measures of vision, could be meta-analysed. The limited evidence from meta-analysis suggests that visual field defects (four studies; RR 1.51 (95%CI 1.23-1.85); p<0.001; I²=46.79%), and contrast sensitivity (two studies; RR 1.40 (95%CI 1.08-1.80); p=0.01, I²=0.11%) and visual acuity loss (five studies; RR 1.21 (95%CI 1.02, 1.43); p=0.03, I²=28.49%) may increase crash risk. The results are more inconclusive for available evidence for associations of glaucoma (five studies, RR 1.27 (95%CI 0.67-2.42); p=0.47; I²=93.48%) and cataract (two studies RR 1.15 (95%CI 0.97-1.36); p=0.11; I²=3.96%) with crashes. Driving cessation may also be linked with glaucoma (XXX)...”

Although only my statistical opinion was requested 1. I am a bit mystified by the statement “This review is part of the Lancet Global Health Commission on Global Eye Health”.

This line has been amended for clarity as follows on page 4:

“This review was completed in collaboration with the Lancet Global Health Commission on Global Eye Health...”

2. The authors report that 1 systematic review was included in their review. Can the authors assure the readers that the studies in the MA were not also included among the 101 individual studies assessed, otherwise would not double counting exist?

The studies in the review by X et al and also detected in the search were cross-checked for duplication and unique studies included. Nonetheless, a note about this has been added to the “Discussion” section, page 13:

“... due to the small number of studies included in each meta-analysis. **The published meta-analysis was summarised narratively to ensure duplicate studies were not included in this evidence synthesis.** Grey literature and non-English studies were not included...”

3. The authors recommend more studies in LMIC but I do wonder if this is really a research priority for these countries with limited resources and numerous other health and safety issues. Why would one expect associations between visual disturbances and motor vehicle crashes to vary between HIC and LMIC?

Thank you for your comment. The authors agree that, although research in LMICs would be helpful, there are likely conflicting priorities. However, it is still important to recognise that motorisation is increasing rapidly in many LMICs with many road fatalities coming from these regions. Further, as access to proper eye health care services in many LMICs remains limited, the effect sizes may be different to HICs. It may therefore be better to position the argument as using evidence from HICs to help guide the creation of future studies in LMICs. Large-scale, resource-intensive studies may not be possible at the moment, but smaller cross-sectional studies can provide more information which future studies can build upon. These thoughts have been added to the “Discussion” section on page 12 and page 13:

Page 12:

“Even though most LMICs do have guidelines on vision, especially for commercial drivers, it is apparent from the studies in this review that many drivers unfortunately do not satisfy these conditions. **This may be because many people in LMICs lack access to eye-health care services.** The evidence for a corresponding increase in MVCs in LMICs is not well established with only one systematic review identified looking at associations between vision and MVC risk from LMIC. **Though data from HICs can inform research and policy development in LMICs, increasing the evidence base from LMICs will ensure that interventions to reduce MVCs and maintain access to driving in LMIC can be reflective of the local context.**”

Page 13:

“However, the current literature is highly heterogeneous, **and more studies are needed from LMICs to ensure that what is known about vision and driving in these settings.** Future studies should aims to address...”

VERSION 2 – REVIEW

REVIEWER	Murdoch, Ian UCL, Institute of Ophthalmology
REVIEW RETURNED	22-Mar-2023

GENERAL COMMENTS	Many thanks. I was particularly impressed with reviewer 6 whose very good points you have addressed carefully. From my point of view the paper adds in that it established the limits of our current knowledge on the topic. I have one further point worth adding to the discussion and that is the issue of co-morbidity. Driving involves more than simply vision and a limitation of missing out on co-morbidity (which many with sight loss have) is going to be another unquantified course of bias.
--

VERSION 2 – AUTHOR RESPONSE

Reviewer: 5

Comments to the Author:

Many thanks. I was particularly impressed with reviewer 6 whose very good points you have addressed carefully. From my point of view the paper adds in that it established the limits of our current knowledge on the topic. I have one further point worth adding to the discussion and that is the issue of co-morbidity. Driving involves more than simply vision and a limitation of missing out on co-morbidity (which many with sight loss have) is going to be another unquantified course of bias.

Thank you for this suggestion. The authors agree with your input and have added this to the Discussion (page 13):

“There are, however, limitations which should be acknowledged... It is therefore imperative that the meta-analysis results be considered alongside the narrative summaries to gain the full picture of the literature in this field. Further, this review did not consider how comorbidities, alongside vision impairment, can impact the risk of crash and driving cessation. Older adults with a vision impairment have been found to be twice as likely than those without a vision impairment to have five or more physical and/or cognitive comorbidities.[1] It is possible that the association with vision is confounded by the impact of comorbidities. Unfortunately, not all the studies included in this review reported on the comorbidities of their participants, limiting our ability to explore this possible source of bias and the extent to how this might have explained the heterogeneity of the pooled estimates via meta-regression. There were great variations...”

New Reference Added:

1. Court H, McLean G, Guthrie B, Mercer SW, Smith DJ. Visual impairment is associated with physical and mental comorbidities in older adults: a cross-sectional study. *BMC medicine*. 2014;12:181.